# Internal Explosion Performance of RDX@Nano-B Composite Explosives

**DOI:** 10.3390/nano13030412

**Published:** 2023-01-19

**Authors:** Peng Xi, Shiyan Sun, Yu Shang, Xiaofeng Wang, Jun Dong, Xuesong Feng

**Affiliations:** 1College of Weaponry Engineering, Naval University of Engineering, Wuhan 430033, China; 2Xi’an Modern Chemistry Research Institute, Xi’an 710065, China; 3School of Aerospace Engineering, Xi’an Jiaotong University, Xi’an 710049, China

**Keywords:** boron nano-powder, boron-containing composites, metalized explosives, internal explosion performance

## Abstract

Boron powder is an additive for metalized explosives with great application potential. To improve the energy release ability of boron powder, the composites of RDX and nano-boron (RDX@Nano-B) were prepared by the spray-drying process, and the metalized explosives based on it were designed (named PBX-B1). The detonation heat and explosion pressure of boron-containing explosives PBX-B1 under vacuum and air conditions were measured and analyzed by an internal explosion test. On the other hand, the equilibrium pressure and energy release of the PBX-B1 explosive system after detonation were analyzed and compared with that of an explosive formulation of the same composition (named PBX-B2). Results showed that the detonation heat of PBX-B1 was 7456 J/g in a vacuum environment, which was 34.8% higher than that of RDX (5530 J/g). However, in the air environment, the detonation heat of PBX-B1 increased by 19.2% compared with that in the vacuum environment, and the explosive gas products mainly included N_2_, NO_x_, CO, H_2_O, CH_4_, HCN, and CO_2_. The peak pressure and equilibrium pressures of PBX-B1 were 11.2 and 0.42 MPa, which were increased by 155% and 75% compared with the vacuum environment, respectively. It is worth noting that, compared with that of PBX-B2, the released energy in the aerobic combustion stage and equilibrium pressure of PBX-B1 were increased by 49.8% and 10.5%. This study demonstrated the strategy of improving the energy release of boron-containing metalized explosives through the design of an explosive microstructure, which provides important clues for the design of higher-energy metalized explosives.

## 1. Introduction

Metalized explosives, which are represented by aluminum-containing explosives and possess the characteristics of high detonation heat, high temperature, and long detonation reaction time, etc., can significantly improve the energy and work capacity of explosives and are widely used in advanced warhead charges, such as warm pressure weapons, penetration weapons, and underwater weapons [1,2,3]. Although aluminum powder has a high calorific value and stability and is widely adopted in explosives, boron powder has a significantly higher volume unit calorific value than aluminum powder from the perspective of thermodynamics. Therefore, it is of great significance to study the application of boron powder in metalized explosives and to further improve the explosion energy and work capacity of explosives [4,5,6].

Although the heat release of boron powder (140 GJ/m^3^) in the process of a combustion and explosion reaction is much higher than that of aluminum powder (85 GJ/m^3^), the ignition and combustion of boron powder are difficult, and the energy release is inhibited. This phenomenon may be caused by the fact that boron powder has a higher boiling point and a lower combustion temperature than Al powder, forming a B_2_O_3_ liquid film on the combustion surface, which wraps around the surface of boron and inhibits the combustion of boron particles [7,8]. To improve the ignition and combustion characteristics of boron particles, it is an effective strategy to improve the chemical reactivity of boron particles by using the advantages of nanomaterials, such as the faster release of chemical energy and a more complete chemical reaction. Compared with micron boron powder, nano-boron particles can shorten the melting time, rapidly remove the B_2_O_3_ liquid film, and finally improve the reactivity of boron particles, which has been confirmed by Sullivan et al. through constant volume combustion tests [9,10,11]. The application of nano-boron particles in explosives is beneficial to exert the energy advantage of boron, but the corresponding trouble is that it is easily agglomerated and loses activity, which limits its application in explosives [12,13]. Therefore, it is important to solve the energy release problem of boron-containing metalized explosives to prepare composite particles of oxidizer and nano-boron using a microstructure design [14,15].

In this paper, the composites of RDX and nano-boron (RDX@Nano-B) were prepared by the spray-drying method. The characteristics of the composites were compared with the mixture of RDX and nano-boron (RDX/Nano-B). The energy contribution of RDX@Nano-B composites to boron-containing metalized explosives was analyzed and the reaction mechanism during the boron implosion process was clarified.

## 2. Experimental Section

### 2.1. Materials

Amorphous boron powder (purity 99%, particle size 10–200 nm, and the content of boron oxide/boric acid less than 0.8%) was prepared by Xi’an Jiaotong University. RDX (purity 99.9%, particle size 1–20 μm) was purchased from Gansu Yinguang Chemical Group Co., LTD.

### 2.2. Fabrication of RDX@Nano-B Composites and RDX/Nano-B Mixture

The RDX@Nano-B composites were prepared by a spray dryer developed by the Xi’an Institute of Modern Chemistry, and the preparation process is shown in Figure 1.

As shown in Figure 1, the RDX was first dissolved in cyclohexanone at room temperature to form a solution, and then the boron powder was added with vigorous stirring to form a homogenous suspension. The composite explosive particles were prepared by mixing the static mixer with nitrogen and then spray drying with the feed rate of 5 mL/min and the air extraction rate of 35 m^3^/h. Then, both the obtained explosive particles and the RDX-saturated solution were added to an oarless mixer and stirred at the rate of 5 r/min for 30 min to repair the explosive surface. Finally, the mixture was filtered and dried to obtain RDX@Nano-B composites.

The RDX/Nano-B mixture was obtained by adding nano-boron and RDX into ethanol under vigorous stirring to form a homogenous suspension, followed by filtering and drying.

### 2.3. Preparation of Boron-Containing PBX Explosive

Two boron-containing polymer-bonded explosives (named PBX) were designed for comparative study. According to [16], the RDX@Nano-B composites and the mixture of RDX/Nano-B were added to the petroleum ether solution. The ethylene-vinyl acetate copolymer (named EVA) was also added as a surfactant. After thoroughly mixing and drying, the boron explosive modeling powder based on the composites (named PBX-B1) and the explosive modeling powder based on the mixture (named PBX-B2) were obtained. Among them, the molding charge column with a diameter of 35 mm was obtained via the mold-pressing method under the pressure of 100 MPa. The content of RDX was 75%, the content of boron powder was 20%, and the content of EVA was 5% in PBX-B1 and PBX-B2. The explosive densities of PBX-B1 and PBX-B2 were 1.71 and 1.70 g/cm^3^, respectively.

### 2.4. Internal Explosion Test

The internal explosion test used a closed calorimetric bomb, in which the outer layer was an annular thermostatic outer barrel containing distilled water, and the inner layer was a calorimetric bomb made of stainless steel. The structure of the calorimetric bomb is illustrated in Figure 2. Boron-containing explosives with a mass of 50 g were detonated in the center of the calorimetric bomb by a #8 detonator. The temperature change of the distilled water before and after the implosion test was obtained by the temperature sensor installed on the outer cylinder, and the total energy of the explosive after detonation was calculated. After deducting the calorific value of the detonator, namely, each detonator is 6552 J, the detonation heat of the explosive was obtained. The explosive detonation heat was calculated by the temperature change of distilled water before and after the implosion test. The calorimetric capacity of the calorimetric bomb was 20 L. On the top of the calorimetric bomb, a pressure sensor, a gas-collecting device, and a gas channel were equipped. The detonation-produced gas was injected or extracted through the gas channel and the gas product was collected by the gas-collection device for chromatographic analysis [17].

In the pressure measurement test, the voltage–time signal was obtained by the pressure sensor, and the high-frequency component of the shock wave was filtered out by the filter module, to realize the direct measurement of the low-equilibrium pressure. The pressure sensor adopted the Kunshan dual-bridge CYG400 high-frequency dynamic piezoresistive pressure sensor with a range of 6 MPa and an accuracy of 0.5. Among them, the response frequency of the pressure sensor was 150 kHz, the natural frequency was 100–500 Hz, and the pressure rise time was 0.24 ms. The temperature sensor adopted WRe5/26 thermocouples, diameter 0.2 mm, measuring range 0~2500 °C, and a response time of less than 10 ms.

The vacuum environment: the air in the bomb body was removed until the pressure was less than 100 Pa, it was filled with nitrogen to 97,200 Pa, and then nitrogen was removed (via vacuum) to 80 Pa. The air environment: internal pressure was 97,200 Pa, and the oxygen content was 20.52%.

## 3. Results and Discussion

### 3.1. Morphology Characterization of the RDX@Nano-B Composites

To intuitively understand the surface morphology of the composites and the mixture, the particle morphology, size, and distribution of boron powder, composites, and the boron-containing explosives were measured by scanning electron microscopy (SEM) tests, and the results are shown in Figure 3.

Nano-boron is amorphous with a particle size of less than 200 nm and an irregular shape. Figure 3a clearly shows that the B-particles adhered to each other. Figure 3b displays the morphology of the obtained RDX@Nano-B composites, which confirmed that the nano-boron was embedded in the RDX crystal and exhibited a uniformly dispersed surface. Figure 3c showed that, with the coating of EVA, RDX@Nano-B composites were directly bonded to each other to form spherical particles with a size of about 1 mm. Figure 3d shows the morphology of the mixture of boron and RDX, which also clearly exhibited an agglomeration structure of nano-boron and RDX. The strong surface adsorption of boron nanoparticles and EVA enhanced their agglomeration. A comparison between Figure 3c and 3d showed that the agglomerated boron number of PBX-B1 composites prepared by the spray-drying process was smaller than that of PBX-B2. On the other hand, the surface morphology of PBX-B1 particles was more regular and smoother.

### 3.2. The Detonation Heat in Different Environments

To characterize the energy contribution of boron to explosives, the detonation heat of boron-containing explosives was tested in vacuum and air conditions. Each group was tested three times, and the average value is shown in Table 1.

As can be seen from Table 1, under a vacuum environment, the average detonation heat of PBX-B1 and PBX-B2 was 7456 and 7346 J/g, respectively, which is 7.2% and 5.6% higher than that of aluminum-containing explosives (6956 J/g), and 34.8% and 32.8% higher than that of RDX (5530 J/g) [18]. On the other hand, it was also confirmed that the detonation heat of the boron-containing PBX-B1 explosive was higher than that of PBX-B2.

A comparison between the detonation heat of PBX-B1 and PBX-B2 in an air environment also showed that the tested value in air conditions was significantly higher than that in a vacuum environment. According to the secondary reaction theory, for metalized explosives, oxygen in the air environment participates in the post-combustion reaction of explosives, which can be regarded as an explosive mixture formed by explosives and air. The oxygen in the air changes the oxygen balance of the original explosive. The oxygen content of the whole system increases with the occurrence of the explosion, which promotes the contact of metal powder and explosive intermedia with oxygen. The complete reaction of metal powder improves, which effectively promotes the energy release of explosives and thus increases the detonation heat of explosives. Compared with the vacuum condition, the detonation heat value of PBX-B1 and PBX-B2 in the air increased by 19.2% and 13.0%, respectively.

### 3.3. Explosion Pressure

The system pressure changes during the detonation process were recorded. The pressure–time curves of PBX-B1 and PBX-B2 in a vacuum and air environment were obtained by the pressure sensor and the results are shown in Figure 4.

Figure 4 reveals that, under vacuum conditions, the pressure curves of PBX-B1 and PBX-B2 were almost the same, and the peak pressure was 4.4 and 4.3 MPa, respectively, and the equilibrium pressure was 0.24 MPa. On the other hand, it was also distinctly observed that, in the air environment, the peak pressure of PBX-B1 and PBX-B2 was 11.2 and 8.7 MPa, respectively, and the equilibrium pressures were 0.42 and 0.38 MPa, respectively. A comparison of the pressure showed that the explosive energy can only be propagated by the generated gas that expands outwards with the occurrence of the reaction. As a result, the explosive pressure rapidly decayed and the balance pressure was low. However, in the air environment, explosion energy can be propagated outward through the air, and the pressure attenuation was slower than that in the vacuum environment. Finally, the peak pressure of PBX-B1 and PBX-B2 was 155% and 102% higher than that in the vacuum environment, respectively.

### 3.4. Gas Product Analysis

The gas products of explosives are directly related to the implosion pressure. By collecting the gas products of boron-containing explosives in the air environment for chromatographic analysis, the possible explosive reaction processes of boron-containing explosives were further speculated. The gas products of PBX-B1 in the air environment were mainly N_2_, NO_x_, CO, H_2_O, CH_4_, HCN, and CO_2_, while the gas products of PBX-B2 were N_2_, NO_x_, CO, H_2_O, and CH_4_. The molarity concentrations of some gas products are shown in Table 2.

The thermal decomposition reactions of RDX mainly included the competitive N-NO_2_ and H_2_C-N bonds breaking at the same time. Under ideal detonation conditions, the main elemental chain of RDX breaks first [19], which results in the formation of N_2_O and CH_2_O. Due to the instability of N_2_O, N_2_ and NO_2_ were further formed. Subsequently, CH_2_O reacted with NO_2_ and formed N_2_, NO, CO, CO_2_, H_2_O, and CH_4_, etc. During the course of boron-containing PBX-B1 explosives, N-NO_2_ fracture occurred in RDX first, and the nitro group was removed. Then, the C-N bond in the main chain was broken and the intermediate of HCN and CH_2_NNO_2_ was generated. Finally, the CH_2_NNO_2_ was further decomposed into N_2_, NO, CO, CO_2_, H_2_O, and CH_4_. Due to the formation of composites between RDX and nano-boron, the reactivity of nano-boron was improved during the decomposition and exothermal process of RDX, which promoted the reaction between nano-boron and CO_2_. Therefore, the content of CO_2_ was greatly reduced, and the content of CO was increased. In conclusion, the composition of the composites affects the reaction process of RDX and eventually leads to the formation of different products.

### 3.5. Explosive Reaction Mechanism of Boron-Containing Composites

The structure of boron and RDX as well as the explosion environment have a certain effect on the energy release of boron-containing explosives in the confined space. According to the classical ZND model of the detonation wave, the theoretical model of intrinsic detonation, and the difference of energy and reaction products of boron-containing explosives in a vacuum and air environment, we assumed that the detonation process of boron-containing explosives is divided into three stages, as shown in Figure 5. The first stage mainly occurred in the anaerobic explosive reaction, in which the released energy (*Q*_1_) was mainly attributed to the molecular reaction, i.e., the decomposition reaction of RDX, which does not require oxygen extraction from the surrounding air. RDX detonation-released energy provides a high-temperature and high-pressure environment for boron, reduces the reaction threshold of boron, and enhances the reactivity of boron. According to the eigenvalue detonation theory [20,21], it is difficult for boron in the mixed structure to be oxidized in the first stage. The RDX@Nano-B reduces the spatial distance between the explosive products and boron, which promotes the formation of boron oxides. The boron of this structure reacts with reactive oxygen atoms (intermediate components formed during the decomposition of RDX) to produce boron oxides.

The second stage is also the anaerobic combustion reaction stage, and the released energy is *Q*_2_. In this stage, there is no external air involved in the reaction. The oxygen-free explosion reaction provides the initial high-temperature environment for the heating and ignition of boron particles, which improves the energy release rate of boron particles, and further promotes the reaction between boron and RDX decomposition products as well as CO_2_, as has been proven through experiments [22,23,24]. The change rate of oxide layer thickness is inversely proportional to the square of the boron particle size. Therefore, the smaller the boron particle size, the faster the change of the thickness of the boron oxide layer, and the easier the boron powder is to be ignited.

The third stage is the aerobic combustion reaction, and the released energy is *Q*_3_, which comes from the reaction of activated boron and air. The comparison of *Q*_1_ + *Q*_2_ and *Q*_3_ of PBX-B1 and PBX-B2 is illustrated in Figure 6.

Figure 6 clearly shows that in the first and second stages, due to the low oxygen content in the boron-containing explosive and the low concentration of CO_2_, the detonation heat of composites was increased by 1.5% (110 J/g) over the mixture. In the third stage, accompanied by the beginning of the reaction of boron with oxygen and CO_2_ in the air environment, the boron conversion rate rapidly increased and the energy increased by nearly 49.8%, which is much higher than that of the conventional mixed structure. The determining factor for boron powder ignition is the ignition of the boron particles [25,26]. When the surface temperature of the boron particles increased enough, the oxidant removal rate of the surficial layer through evaporation and surface chemical reactions was greatly increased. With the heating of particles by the anaerobic explosion and combustion reaction, the ignition time of boron powder reduced, and the oxidation rate of boron powder improved. All the changes occurred quickly with the proceeding of the oxidation reaction, which improved the oxidation rate of boron and increased the consumption rate of oxygen and CO_2_ in the air. In summary, compared with the mixture’s explosive structure, the RDX@Nano-B explosive composites’ structure may not only shorten the spatial distance between the explosive products and boron, but may also increase the degree of mixing with oxidizing substances in the three stages of the explosive reaction, which will promote the formation of boron oxides, finally increasing the total energy by about 7%.

## 4. Conclusions

To improve the energy release ability of boron powder in explosives, the RDX@Nano-B composites with nano-boron embedded in RDX crystal were prepared by the spray-drying method. The boron-containing explosive PBX-B1 was designed. The detonation heat and pressure of PBX-B1 were studied by the internal explosion test under both vacuum and air conditions. Based on the comparison of PBX-B1 and PBX-B2, the following conclusions could be drawn:

(1) The PBX-B1 explosives had a slightly higher detonation heat than PBX-B2. Under vacuum conditions, the detonation heat of PBX-B1 was 7456 J/g, which was higher than that of PBX-B2.

(2) In the air environment, the explosive peak pressures of PBX-B1 and PBX-B2 were 11.2 and 8.7 MPa, respectively, which were 155% and 102% higher than those in the vacuum environment. In addition, the equilibrium pressures were 0.42 and 0.38 MPa, respectively, which were 75% and 58% higher than those in the vacuum environment. The equilibrium pressure of PBX-B1 was 10.5% higher than that of PBX-B2.

(3) The detonation process of boron-containing explosives was mainly composed of three stages: anaerobic detonation, anaerobic combustion, and aerobic combustion. In the anaerobic detonation and anaerobic combustion stages, the detonation heat of PBX-B1 was increased by 1.5% compared with PBX-B2, while the released energy of PBX-B2 increased by nearly 49.8% compared with that of PBX-B2 in the aerobic combustion stage. The introduction of B to PBX increases the energy density and the reactivity of explosives.

## Figures and Tables

**Figure 1 nanomaterials-13-00412-f001:**
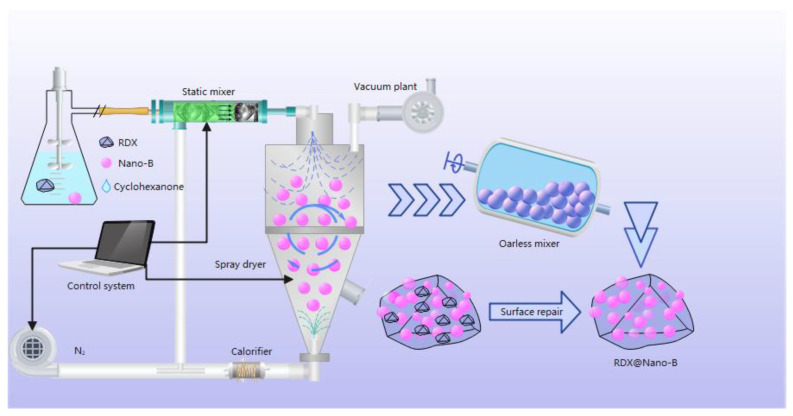
The preparation process of complex RDX@Nano-B.

**Figure 2 nanomaterials-13-00412-f002:**
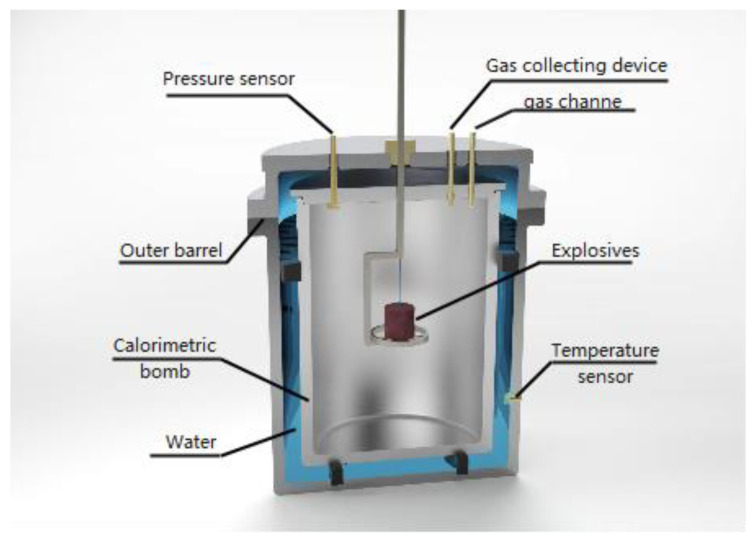
Schematic illustration of a calorimetric bomb.

**Figure 3 nanomaterials-13-00412-f003:**
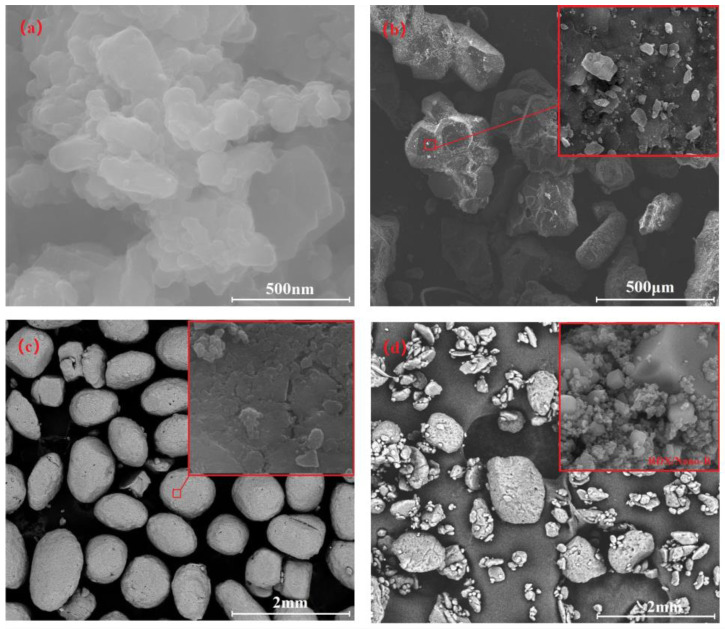
SEM morphology of Nano-B (**a**), RDX@Nano-B composites (**b**), PBX-B1 (**c**), and PBX-B2 (**d**) explosives.

**Figure 4 nanomaterials-13-00412-f004:**
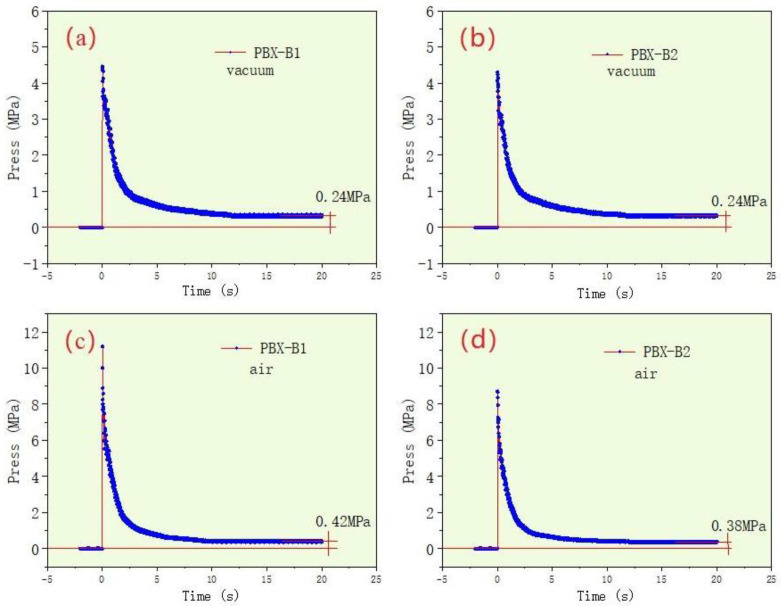
The pressure–time curves of PBX-B1 and PBX-B2 under vacuum (**a**,**b**) and air (**c**,**d**) conditions, respectively.

**Figure 5 nanomaterials-13-00412-f005:**
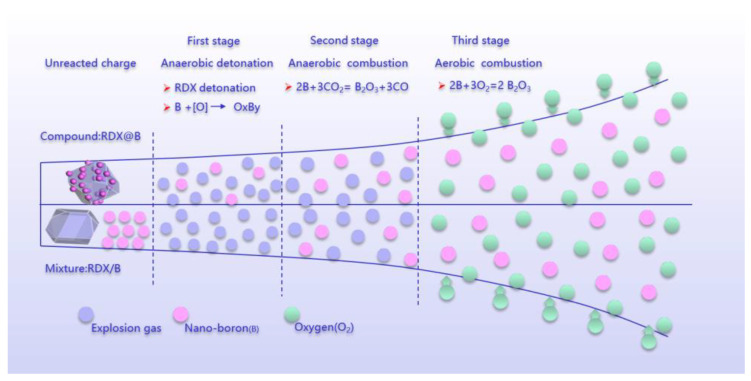
Illustration of implosion reaction diagram.

**Figure 6 nanomaterials-13-00412-f006:**
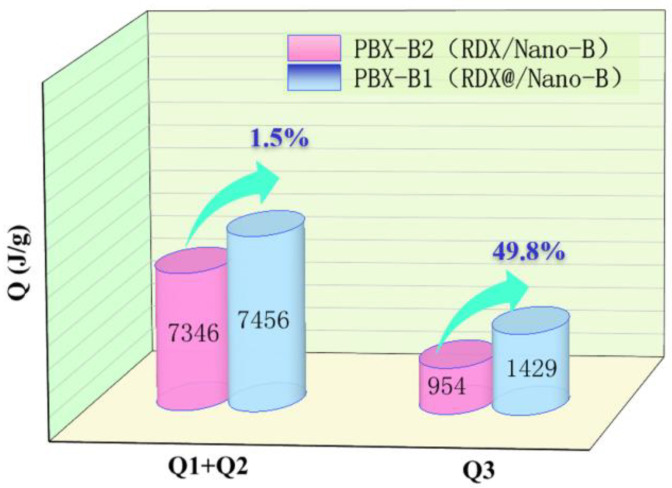
Schematic diagram of energy increment of boron explosives.

**Table 1 nanomaterials-13-00412-t001:** Detonation heat values of boron-containing explosives.

Samples	*Q*_Vacuum_ (J/g)		*Q*_Air_ (J/g)	
	Experimental	Average	Experimental	Average
PBX-B1	7432	7456	8866	8885
	7448		8777	
	7488		9012	
PBX-B2	7387	7346	8229	8300
	7322		8357	
	7329		8314	

**Table 2 nanomaterials-13-00412-t002:** Gaseous products of explosion in the air condition.

Explosives	CO/%	CH_4_/%	HCN/%	CO_2_/%
PBX-B1	28.971	3.588	3.732	0
PBX-B2	25.147	2.169	0	2.793

## Data Availability

Data that support the findings of this study are included within the article.

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
