# Peer review of "Internal Explosion Performance of RDX@Nano-B Composite Explosives"

_nanomaterials, 2023, doi:10.3390/nano13030412_

Round 1
Reviewer 1 Report
The manuscript contains the results of measurements of the heat of explosion of mixtures of nanosized boron with RDX. The results showed a significant increase in energy release and pressure due to the introduction of boron. However, the text of the manuscript has significant shortcomings.
1. Line 59. "In this paper, the composites of ammonium nitrate oxidizer, RDX, and nano-boron (RDX@Nano-B) were prepared by spray drying method" - ammonium nitrate is not mentioned anywhere else in the manuscript
2. Nanosized boron powder was used, but impurity analysis and boron oxide content are not specified. An additional analysis of the actual purity of the product should be carried out.
3. The manuscript does not indicate the concentrations of the components in the compositions and the charge density. This makes it impossible to compare the values of the parameters given in the work with the data known from the literature.
4. The authors compare the heat of explosion of compositions with boron and aluminum. According to their conclusion, compositions with B are superior in terms of heat of explosion to a mixture with Al (when tested in vacuum). But, again, at what concentration of additives a comparison is made? The fact is that the maximum values of the heat of explosion are achieved at different contents of B and Al (if only because of the difference in their atomic weights). According to the calculations, it follows that there are concentration ranges within which, indeed, compositions with B are superior in terms of the heat of explosion to compositions with Al. However, at the optimum concentration (for B it is noticeably lower), the maximum value of the heat of explosion is higher in the case of Al.
5. The authors allow some manipulation when comparing the obtained values. So, in fig. 6 shows a gain in energy of 49.8%. Impressive! But this is when comparing the growth values due to the reaction with air. If we consider the total heat release, then the excess is only ~ 7%. However, the effect due to the improvement of the structure is still there.
6. Paragraph 3.4 states that H2O is formed in the RDX explosion products. However, there is no further mention of water. Only the reaction of B with CO2 and the reaction of B with atmospheric oxygen are considered. Whether B interacts with H2O is not clear from the text.
7. Table 2 shows the gas composition of the products, which add up to approximately 30%. What is the composition of other gases? Where is nitrogen and its oxides?
8. Figure 5. At the first stage, the reaction B + O> OxBy is indicated, it is not clear what is the source of atomic oxygen at this stage?
9. Based on the results of pressure measurements, the results of an explosion in vacuum and in air are compared. This comparison is not entirely correct, since a shock wave forms in air, but not in a vacuum. It would be more correct to compare the pressure during an explosion in air and an inert gas (nitrogen, argon).

Author Response
Dear Reviewer:
Thank you for your good comments and suggestions on our paper entitled “Internal explosion performance of RDX@Nano-B composite explosives” (Manuscript ID: nanomaterials-2135890). We have carefully revised the manuscript according to your comments. We acknowledge your comments and suggestions, which are valuable in improving the quality of our work. Corresponding revisions have been made and highlighted with red in the manuscript. The responses to your comments one point to one point are attached as follows in the email. Please check them again. If you have any further questions concerning the manuscript, please contact me without hesitation. Thanks for your work on our manuscript again.
Best,
Shiyan Sun
Email: 415088546@qq.com
College of Weaponry Engineering,
Naval University of Engineering,
Wuhan 430033, China.
Responses to the referees
Reviewer: 1
The manuscript contains the results of measurements of the heat of explosion of mixtures of nanosized boron with RDX. The results showed a significant increase in energy release and pressure due to the introduction of boron. However, the text of the manuscript has significant short comings.
Q1:Line 59. "In this paper, the composites of ammonium nitrate oxidizer, RDX, and nano-boron (RDX@Nano-B) were prepared by spray drying method" - ammonium nitrate is not mentioned anywhere else in the manuscript.
Re: Thanks for the reviewer’s constructive comments. The suggestion is accepted. There is no ammonium nitrate in the explosive formula material. We have revised the corresponding content in the manuscript Page 2 Line 59-60.
Page 2 Line 59-60:
In this paper, the composites of RDX, and nano-boron (RDX@Nano-B) were pre-pared by spray drying method.
Q2:Nanosized boron powder was used, but impurity analysis and boron oxide content are not specified. An additional analysis of the actual purity of the product should be carried out.
Re: Thanks for the reviewer’s careful examination. The suggestion is accepted. The boron in this work is the highly active nanosized boron powder that obtained from Xi'an Jiaotong University. The purity of the boron powder is 99.9% and the oxygen content is 0.07%, which were examined by scanning electron microscope energy dispersive spectrum technology. The obtained results are shown in table 1, and the morphology of nanosized boron powder is shown in Figure 1.
Table 1 The SEM energy spectrum analysis data.
Element |
B |
Fe |
Ca |
Mg |
Cu |
Mn |
Na |
Co |
Weight (%) |
99.9 |
0.001 |
0.001 |
0.002 |
0.001 |
0.001 |
0.003 |
0.001 |
Element |
Zn |
Ni |
Pb |
K |
N |
C |
S |
0 |
Weight (%) |
0.001 |
0.002 |
0.001 |
0.001 |
0.004 |
0.001 |
0.001 |
0.07 |
Figure 1. The SEM morphology of nanosized boron powder.
In addition, we also used the chemical titration method to determine the boron content (the titration principle is as follows), the results show that the content of elemental boron is not less than 99%, and the content of boron oxide/boric acid is less than 0.8%. We have revised the corresponding content in the manuscript Page 2 Line 66-67.
B2O3(s)+3H2O (l)=2H3BO3 (1)
2CH2(OH)(CHOH)2CH2OH+H3BO3={[CH2(OH)]2C2H202BO2C2H2[CH2(OH)]2}-+H+ +3H2O (2)
H+ +OH- = H2O (3)
Page 2 Line 66-67:
Amorphous boron powder (purity 99%, particle size 10-200 nm, and the content of boron oxide/boric acid is less than 0.8%) was prepared by Xi 'an Jiaotong University.
Q3:The manuscript does not indicate the concentrations of the components in the compositions and the charge density. This makes it impossible to compare the values of the parameters given in the work with the data known from the literature.
Re: Thanks for the reviewer’s constructive comments. The suggestion is accepted. It is important to determine the concentration of components in explosive compositions and charge density for the analysis of the explosive reaction mechanism. We have revised the corresponding content in the manuscript Page 3 Line 95-96.
Page 3 Line 95-96:
The content of RDX is 75%, the content of boron powder is 20% and the content of EVA is 5% in PBX-B1 and PBX-B2. The explosive densities of PBX-B1 and PBX-B2 are 1.71g/cm3 and 1.70g/cm3, respectively.
Q4:The authors compare the heat of the explosion of compositions with boron and aluminum. According to their conclusion, compositions with B are superior in terms of heat of explosion to a mixture with Al (when tested in a vacuum). But, again, at what concentration of additives a comparison is made? The fact is that the maximum values of the heat of explosion are achieved at different contents of B and Al (if only because of the difference in their atomic weights). According to the calculations, it follows that there are concentration ranges within which, indeed, compositions with B are superior in terms of the heat of explosion to compositions with Al. However, at the optimum concentration (for B it is noticeably lower), the maximum value of the heat of explosion is higher in the case of Al.
Re: Thanks for the constructive comments. In Section 3.2 of this manuscript, the detonation heat of two kinds of boron-containing explosives is analyzed and compared with RDX and aluminum-containing explosives of the same proportion. Because the molecular weight of boron is smaller than that of aluminum, boron powder consumes significantly more oxygen than aluminum, which is one of the reasons for the high calorific value per unit mass of boron. Metalized explosives require different metal powder contents under different working conditions (implosion, air explosion, underwater explosion). The aluminum powder content of implosion explosives is generally 35%~45%, and the maximum content of boron powder is more than 20% theoretically calculated. The comparison of the data in this work is mainly to express the difference in the detonation heat of the two structures. The comparison of 20% metal powder with RDX and aluminum-containing explosives does not reach the maximum of the implosion condition. The comparison only provides technical references for experts in the same industry.
Q5:The authors allow some manipulation when comparing the obtained values. So, in fig. 6 shows a gain in energy of 49.8%. Impressive! But this is when comparing the growth values due to the reaction with air. If we consider the total heat release, then the excess is only ~ 7%. However, the effect due to the improvement of the structure is still there.
Re: Thanks for the careful examination and constructive comments. The suggestion is accepted. It could be seen that the reviewer is expertized in this field. Based on the speculation of the explosive reaction mechanism of boron-containing explosives, the initiation process of boron-containing explosives is divided into three stages. The first stage and the second stage are the anaerobic explosion reaction and the anaerobic combustion reaction after the explosion, respectively. The energy released (Q1+Q2) is mainly the reaction within the molecular compounds of high-energy explosive (such as the decomposition reaction of RDX) and the reaction of boron with oxidizing intermediates and CO2. Due to the low concentration of the oxidizing intermediates and CO2 in these two stages, the increase in the detonation heat ratio of the composite to the mixture is small, about 1.5% (110 J/g). The third stage is the aerobic combustion reaction stage after the explosion, and the energy released by the reaction of activated boron with air is denoted as Q3. In the third stage, accompanied by the beginning of the reaction of boron with oxygen and CO2 in the air environment, the boron conversion rate increases rapidly and the energy increases by nearly 49.8% (475 J/g), which is much higher than that of the conventional mixed structure. In summary, compared with the mixture explosive structure, RDX@Nano-B explosive composites structure may not only shorten the spatial distance between the explosive products and boron, but also increase the degree of mixing with oxidizing substances in the three stages of the explosive reaction, which will promote the formation of boron oxides, and finally increasing the total energy by about 7%. We have revised the manuscript according to the comment on Page 8 Line 248-253.
Page 8 Line 248-253:
In summary, compared with the mixture explosive structure, RDX@Nano-B explosive composites structure may not only shorten the spatial distance between the explosive products and boron, but also increase the degree of mixing with oxidizing substances in the three stages of the explosive reaction, which will promote the formation of boron oxides, and finally increasing the total energy by about 7%.
Q6:Paragraph 3.4 states that H2O is formed in the RDX explosion products. However, there is no further mention of water. Only the reaction of B with CO2 and the reaction of B with atmospheric oxygen are considered. Whether B interacts with H2O is not clear from the text.
Re: Thanks for the constructive comments. We have reviewed the related references and reconsidered the reaction between H2O and boron particles. In theory, boron can react with water. But no obvious reaction evidence of boron and water was found in the internal explosion tests in this work. The gaseous H2O was found in the products of both RDX@Nano-B explosive composites and the RDX mixture. At the same time, the thermal decomposition of the boron and RDX mixture was investigated by DSC tests. The obtained results could not prove the obvious reaction between boron and H2O in the explosive gas product of RDX.
Q7:Table 2 shows the gas composition of the products, which add up to approximately 30%. What is the composition of other gases? Where is nitrogen and its oxides?
Re: Thanks for the reviewer’s careful examination and constructive comments. The suggestion is accepted. In this work, there are a lot of explosive reaction gas products of the two kinds of boron-containing explosives. We have supplemented the gaseous products of PBX-B1 and PBX-B2 under ambient air conditions in section 3.4 of the revised manuscript. However, the main difference between the explosive reaction of the two kinds of explosives lies in the four gases described in the paper, such as CO, CO2, HCN, and CH4. We have revised the manuscript according to the comments on Page 6 Line 188-191.
Page6 Line 188-191:
The gas products of PBX-B1 in the air environment are mainly N2, NOx, CO, H2O, CH4, HCN, and CO2, while the gas products of PBX-B2 are N2, NOx, CO, H2O, and CH4. The molarity concentrations of some gas products are shown in Table 2.
Q8:Figure 5. At the first stage, the reaction B + O> OxBy is indicated, it is not clear what is the source of atomic oxygen at this stage?
Re: Thanks for the reviewer’s constructive comments. The suggestion is accepted. The source of oxygen atoms should be attributed to the intermediate components that formed during the decomposition process of organic explosives such as RDX. We have revised and confirmed in Figure 5 of the revised manuscript.
Page 7 Line 220.
Figure 5. Illustration of implosion reaction diagram.
Q9:Based on the results of pressure measurements, the results of an explosion in vacuum and in air are compared. This comparison is not entirely correct, since a shock wave forms in air, but not in a vacuum. It would be more correct to compare the pressure during an explosion in air and an inert gas (nitrogen, argon).
Re: Thanks for the reviewer’s constructive comments. In fact, we have tested the equilibrium pressure of PBX-B1 and PBX-B2 in the nitrogen environment, and the results are shown in Table 2. In the nitrogen environment, the equilibrium pressure is 0.08 MPa higher than that in the vacuum environment, which is smaller than the initial pressure of nitrogen of 0.097 MPa which should be attributed to the reaction between boron and nitrogen. Due to the complex reaction between boron and nitrogen, the limited data in this paper cannot support the research on the reaction mechanism of boron and nitrogen. However, this limitation does not affect the energy comparison between the two kinds of structure in this work.
Table 2 The equilibrium pressure of PBX-B1 and PBX-B2 in different environment conditions.
Samples
Pressure (MPa) |
PBX-B1 |
PBX-2 |
PVacuum |
0.24 |
0.24 |
PN2 |
0.32 |
0.32 |
PAir |
0.42 |
0.38 |

Reviewer 2 Report
The article lacks a lot of information about the experiment.
Please refer to the notes.

Author Response
Dear Reviewer:
Thank you for your good comments and suggestions on our paper entitled “Internal explosion performance of RDX@Nano-B composite explosives” (Manuscript ID: nanomaterials-2135890). We have carefully revised the manuscript according to your comments. We acknowledge your comments and suggestions, which are valuable in improving the quality of our work. Corresponding revisions have been made and highlighted with red in the manuscript. The responses to your comments one point to one point are attached as follows in the email. Please check them again. If you have any further questions concerning the manuscript, please contact me without hesitation. Thanks for your work on our manuscript again.
Best,
Shiyan Sun
Email: 415088546@qq.com
College of Weaponry Engineering,
Naval University of Engineering,
Wuhan 430033, China.
Responses to the referees
Reviewer: 2
Q1:A) There is no information about the following:
1) what is the content of boron additive,
Re: Thanks for the reviewer’s careful examination. The suggestion is accepted. We have supplied related information on Page 3 Line 93-96.
Page 3 Line 93-96:
The content of RDX in PBX-B1 and PBX-B2 is 75%, the content of boron powder is 20%, and the content of EVA is 5%, which have been added in Section 2.3 of the revised manuscript.
2) what is the density of the charges prepared for testing the heat of the explosion,
Re: Thanks for the reviewer’s careful examination. The suggestion is accepted. We have supplied related information on Page 3 Line 96-97.
Page 3 Line 96-97:
The explosive densities of PBX-B1 and PBX-B2 are 1.71 g/cm3 and 1.70 g/cm3, respectively.
3) how charges were prepared,
Re: Thanks for the reviewer’s careful examination. The suggestion is accepted. After the explosive modeling powder is obtained, it is pressed into shape via mould pressing method under the pressure of 100 MPa, and the molding charge column with a diameter of 35 mm is obtained, which have been added in Section 2.3 of the revised manuscript. We have supplied related information on Page 3 Line 93-94.
Page 3 Line 93-94:
Among them, the molding charge column with a diameter of 35 mm is obtained via the mould pressing method under the pressure of 100 MPa.
4) what are the parameters of detonation,
Re: Thanks for the reviewer’s careful examination. The suggestion is accepted. The suggestion is accepted. According to your opinion, the explosive test condition parameters, such as charge mass, initiation mode, pressure sensor, and temperature sensor, have been added in Section 2.4 of the revised manuscript. We have supplied related information on Page 3-4 Line 111-118.
Page 3-4 Line 111-118:
In the pressure measurement test, the voltage-time signal is obtained by the pressure sensor, and the high-frequency component of the shock wave is filtered out by the filter module, to realize the direct measurement of the low equilibrium pressure. The pressure sensor adopts Kunshan dual-bridge CYG400 high frequency dynamic piezo-resistive pressure sensor with a range of 6 MPa and an accuracy of 0.5. Among them, the response frequency of the pressure sensor is 150 kHz, the natural frequency is 100-500 Hz, and the pressure rise time is 0.24 ms. Temperature sensor adopts WRe5/26 thermocouples, diameter 0.2 mm, measuring range 0~2500 ℃, response time less than 10 ms.
5) what was the vacuum in the calorimeter,
Re: Thanks for the reviewer’s careful examination. The suggestion is accepted. The suggestion is accepted. According to your opinion, we have supplied related information on Page 4 Line 119-121.
Page 4 Line 119-121:
The vacuum environment: remove the air in the bomb body until the pressure is less than 100 Pa, fill with nitrogen to 97200 Pa, and then remove nitrogen (vacuum) to 80 Pa. The air environment: internal pressure is 97200 Pa, and the oxygen content is 20.52%.
6) whether the calorimeter was purged with inert gas (what kind) and then removed, or just air was removed,
Re: Thanks for the reviewer’s careful examination. The suggestion is accepted. The calorimeter removes air from the bomb to a pressure of less than 100 Pa, fills it with nitrogen to 97200 Pa, and removes nitrogen (vacuumed) to 80 Pa. We have supplied related information on Page 4 Line 119-121.
Page 4 Line 119-121:
The vacuum environment: remove the air in the bomb body until the pressure is less than 100 Pa, fill with nitrogen to 97200 Pa, and then remove nitrogen (vacuum) to 80 Pa.
7) The heat of detonation of the tested mixtures is compared to the heat of the RDX and RDX/aluminum mixture (lines 132-134) - the source of this result is not indicated.
Re: Thanks for the reviewer’s constructive comment. The suggestion is accepted. The reference literature of data sources for the detonation heat of the RDX and RDX/aluminum mixture have been added in the corresponding position of the revised manuscript. We have supplied revised the related content on Page 5 Line 148-151 and Page 9 Line 313-314.
Page 5 Line 148-151:
As can be seen from Table 1, under a vacuum environment, the average detonation heat of PBX-B1 and PBX-B2 are 7456 J/g and 7346 J/g, respectively, which is 7.2% and 5.6% higher than that of aluminum-containing explosives (6956 J/g), and 34.8% and 32.8% higher than that of RDX (5530 J/g)[18].
Page 9 Line 313-314:
- Xiang, D.; Rong, J.; Li, J.; Feng, X.; Wang, H. Effect of Al/O Ratio on Detonation Performance and Underwater Explosion of RDX-based Aluminized Explosive. Acta Armamentarii. 2013, 34, 45-50.
8) what sensor was used to measure pressure/overpressure?
Re: Thanks for the reviewer’s careful examination. The suggestion is accepted. According to your suggestion, the pressure sensor-related test condition parameters have been added in Section 2.4 of the revised manuscript. We have supplied related information on Page 3-4 Line 111-118.
Page 3-4 Line 111-118:
In the pressure measurement test, the voltage-time signal is obtained by the pressure sensor, and the high-frequency component of the shock wave is filtered out by the filter module, to realize the direct measurement of the low equilibrium pressure. The pressure sensor adopts Kunshan dual-bridge CYG400 high frequency dynamic piezo-resistive pressure sensor with a range of 6 MPa and an accuracy of 0.5. Among them, the response frequency of the pressure sensor is 150 kHz, the natural frequency is 100-500 Hz, and the pressure rise time is 0.24 ms. Temperature sensor adopts WRe5/26 thermocouples, diameter 0.2 mm, measuring range 0~2500 ℃, response time less than 10 ms.
9) What was the initial pressure of air in chamber?
Re: Thanks for the reviewer’s careful examination. The suggestion is accepted. According to your opinion, we have supplied related information on Page 4 Line 119-121.
Page 4 Line 119-121:
The vacuum environment: remove the air in the bomb body until the pressure is less than 100 Pa, fill with nitrogen to 97200 Pa, and then remove nitrogen (vacuum) to 80 Pa. The air environment: internal pressure is 97200 Pa, and the oxygen content is 20.52%.
Q2:B) I have doubts about measuring the heat of detonation of free charges in a vacuum. In Ornellas' research [1], it was proved that the heat obtained in the calorimetric measurement for the charge in heavy confinement and a vacuum corresponds best with the detonation heat. Such conditions can also be obtained in the case of measurements in a calorimetric bomb filled with compressed inert gas [2-5]. Under the measurement conditions presented in the article, the composition of the explosion products changes compared to CJ composition and the measurement result may significantly differ from the detonation heat.
[1]. D. L. Ornellas, Calorimetric Determinations of the Heat and Products of Detonation of Explosives, UCRL-52821, Lawrence Livermore National Laboratory, Livermore 1982.
[2]. F. Volk, J. Energ. Mater., 4 (1986) 93.
[3]. F. Volk and F. Schedlbauer, Propellants, Explosives, Pyrotechnics, 18 (1993) 332.
[4]. F. Volk, Propellants, Explosives, Pyrotechnics, 21 (1996) 155.
[5]. Trzciński, W., Cudziło, S. and Paszula, J. (2007), Studies of Free Field and Confined Explosions of Aluminium Enriched RDX Compositions. Propellants, Explosives, Pyrotechnics, 32: 502-508.
Re: Thanks for the constructive comments. According to the comparison of references, under the inert gas constraint condition (2 MPa), the compressed inert gas also prevents the secondary reaction in the detonation product, which resulted in the detonation calorific value is lower than that under the vacuum condition. In this work, the pressure is not high enough to inhibit the boron reaction under air environment conditions (0.1 MPa) in the Q2 stage, which promotes the oxygen and carbon dioxide in the air directly taken to participate in the secondary reaction of boron (i.e., Q3 stage) and eventually increases calorific value.
Q3: C) Figure 4 - Pressure changes should be represented on a scatter plot with smoothed lines.
Re: Thanks for the reviewer’s careful examination. The suggestion is accepted. According to your opinion, the pressure changes have been represented on a scatter plot with smoothed lines in Figure 4.
Page 6 Line 181-183:
Figure 4. The pressure-time curves of PBX-B1 and PBX-B2 under vacuum (a and b) and air (c and d) conditions, respectively.

Round 2
Reviewer 1 Report
The authors took into account the comments made in the first review of the manuscript and made significant changes to the text. However, the manuscript would still be more understandable if the authors explained the various stages of the boron reaction in more detail. What are the grounds for proving the existence of the first two separate stages? Why does boron react in the second stage only with CO2? However, the authors are entitled to their assumptions, although they do not have sufficient justification. The article can be published without significant corrections.
Author Response
Dear Reviewer:
Thank you for your good comments and suggestions on our paper entitled “Internal explosion performance of RDX@Nano-B composite explosives” (Manuscript ID: nanomaterials-2135890). We have carefully revised the manuscript according to your comments. We acknowledge your comments and suggestions, which are valuable in improving the quality of our work. Corresponding revisions have been made and highlighted with red in the manuscript. The responses to your comments one point to one point are attached as follows in the email. Please check them again. If you have any further questions concerning the manuscript, please contact me without hesitation. Thanks for your work on our manuscript again.
Best,
Shiyan Sun
Email: 415088546@qq.com
College of Weaponry Engineering,
Naval University of Engineering,
Wuhan 430033, China.
Responses to the referees
Reviewer: 1
The authors took into account the comments made in the first review of the manuscript and made significant changes to the text. However, the manuscript would still be more understandable if the authors explained the various stages of the boron reaction in more detail. What are the grounds for proving the existence of the first two separate stages? Why does boron react in the second stage only with CO2? However, the authors are entitled to their assumptions, although they do not have sufficient justification. The article can be published without significant corrections.
Re: Thanks for the reviewer’s constructive comments. The suggestion is accepted. As a typical metallized explosive, the boron-containing explosive has the same or similar detonation reaction process as aluminum-containing explosive to a large extent. The ZND model is a physical model with the highest degree of recognition for the detonation process of aluminum-containing explosives in the industry at present, and its process is shown in the Figure 1 below. Based on the ZND model, the detonation reaction zone is the anaerobic detonation (Q1) of explosives, and the expansion zone of detonation reaction is the redox reaction between metal powder and explosives, also known as the anaerobic combustion reaction (Q2). As one of the most important theories in the industry, the eigenvalue detonation theory model further modifies the ZND model, and holds that metal powder with composite structure can react in the detonation reaction zone, which has been proved by experiments. This work attempts to analyze the reaction process of boron-containing explosives in air environment by referring to the theoretical model of intrinsic detonation. Due to the limited data in this work, the boundary of Q1 and Q2 cannot be accurately defined, so it is assumed that these two processes exist. However, in terms of data analysis, Q1 and Q2 are combined, which is considered as Q1+Q2 stage. According to your suggestions, we further revise the paper to explain the various stages of boron reaction in more detail. In addition, we have limited data to show that boron reacts with carbon dioxide in the second stage. Although there is literature analysis on the possibility of boron reacting with water and nitrogen under certain conditions, it is a pity that the reaction between boron and water and nitrogen cannot be fully proved from our limited data. More importantly, the detonation heat test results in vacuum environment show that there is still a large amount of nitrogen and water after detonation of boron explosives even without the participation of air. We have revised the corresponding content in the manuscript.
Figure 1 The ZND model of detonation wave of aluminum-containing explosives
Page 7 Line 215-218:
According to the classical ZND model of detonation wave, the theoretical model of intrinsic detonation and the difference of energy and reaction products of boron-containing explosives in vacuum and air environment, we assume that detonation process of boron-containing explosives is divided into three stages, as shown in Figure 5.
Page 7 Line 227-230:
RDX@Nano-B reduces the spatial distance between the explosive products and boron and promotes the formation of boron oxides. Boron of this structure reacts with reactive oxygen atoms (intermediate components formed during the decomposition of RDX) to produce boron oxides.
Page 7 Line 234-238:
In this stage, there is no external air involved in the reaction. The oxygen-free explosion reaction provides the initial high-temperature environment for the heating and ignition of boron particles, which improves the energy release rate of boron particles, and further promotes the reaction between boron and RDX decomposition products as well as CO2 has been proved through experiments.

Reviewer 2 Report
Thank you for considering my comments.
Please note that the detonation of charges occurs under different pressures in the calorimeter (in a vacuum and air), so it may affect the measurement results. The contribution of the detonator explosion energy is not described. It must be different for the atmospheres used in the calorimeter - has this effect been considered in the calculations?
Please describe the detonator's contribution to the total energy.
Author Response
Dear Reviewer:
Thank you for your good comments and suggestions on our paper entitled “Internal explosion performance of RDX@Nano-B composite explosives” (Manuscript ID: nanomaterials-2135890). We have carefully revised the manuscript according to your comments. We acknowledge your comments and suggestions, which are valuable in improving the quality of our work. Corresponding revisions have been made and highlighted in red in the manuscript. The responses to your comments one point to one point are attached as follows in the email. Please check them again. If you have any further questions concerning the manuscript, please contact me without hesitation. Thanks for your work on our manuscript again.
Best,
Shiyan Sun
Email: 415088546@qq.com
College of Weaponry Engineering,
The Naval University of Engineering,
Wuhan 430033, China.
Responses to the referees
Reviewer: 2
Thank you for considering my comments.
Please note that the detonation of charges occurs under different pressures in the calorimeter (in a vacuum and air), so it may affect the measurement results. The contribution of the detonator explosion energy is not described. It must be different for the atmospheres used in the calorimeter has this effect been considered in the calculations?
Please describe the detonator's contribution to the total energy.
Re: Thanks for the reviewer’s constructive comments. The suggestion is accepted. According to your modification comments, we further provide the test data. In the internal explosion test, the boron-containing explosives with a mass of 50 g are detonated in the center of the calorimetric bomb with a # 8 detonator. The temperature change of the distilled water before and after the implosion test was obtained by the temperature sensor installed on the outer cylinder, and the total energy of the explosive after detonation was calculated. After deducting the calorific value of the detonator, namely, each detonator is 6552 J, the detonation heat of the explosive is obtained. We have revised the corresponding content in the manuscript on page 3 Lines 103-107.
Page 3 Line 103-107:
The temperature change of the distilled water before and after the implosion test was obtained by the temperature sensor installed on the outer cylinder, and the total energy of the explosive after detonation was calculated. After deducting the calorific value of the detonator, namely, each detonator is 6552 J, the detonation heat of the explosive is obtained.